# Peer review of "Electrochemotherapy for Colorectal Liver Metastasis: What Interventional Radiologists Need to Know"

_livers, doi:10.3390/livers5010006_

Round 1

Reviewer 1 Report

Comments and Suggestions for Authors

The manuscript (review) is very interesting, and will be very useful for researchers and health professionals working in liver and colorectal cancer. The manuscript is well written and has an adequate structure. However, I have some (minor) changes that will improve the scientific quality of the review.

I. Comments

1. Improve the wording of the objective of the review

2. Include a section on the criteria used to select the cited papers (methodology).

3. Include a paragraph with more information on the epidemiology of liver and colorectal cancer, as well as metastasis.

4. Add one or two figures or diagrams showing the main aspects discussed in the review.

5. Highlight the idea of ​​research and clinical projections regarding electrochemotherapy for colorectal liver metastasis

Author Response

Comment 1:

The manuscript (review) is very interesting, and will be very useful for researchers and health professionals working in liver and colorectal cancer. The manuscript is well written and has an adequate structure. However, I have some (minor) changes that will improve the scientific quality of the review.

I.Comments

  1. Improve the wording of the objective of the review
  2. Include a section on the criteria used to select the cited papers (methodology).
  3. 3. Include a paragraph with more information on the epidemiology of liver and colorectal cancer, as well as metastasis.
  4. Add one or two figures or diagrams showing the main aspects discussed in the review.
  5. Highlight the idea of ​​research and clinical projections regarding electrochemotherapy for colorectal liver metastasis

Reply 1:

Dear reviewer, thank you for your precious feedback.

1- The wording has been improved as sentences “78-80” was added in the introduction.

2-As requested, a section “1.1-Methodology” has been added.

3-More information on the epidemiology of CRC and its metastasis was included in lines “31-39, 45-48, 89-91”.

4-“Highlight the idea of ​​research and clinical projections regarding electrochemotherapy for colorectal liver metastasis” was already implemented in points 1 and 2.

5-Figures were added as requested.

Reviewer 2 Report

Comments and Suggestions for Authors

It is an interesting manuscript about “Electrochemotherapy for Colorectal Liver Metastasis: what interventional radiologists need to know”.

*My concern is determined in the following points.

A potential alternative nonsurgical intervention such as electrochemotherapy (ECT) in treatment of CRC oligometastatic liver disease. ECT has been largely used for the treatment of cutaneous and subcutaneous lesions, while its visceral use is currently a novel approach. ECT consists in the administration of intravenous anticancer drugs, followed by the application of intralesional electrode-needles which release localized electrical pulses to induce electroporation, a process that transiently increases cell membrane permeability, thereby facilitating the intracellular delivery of otherwise membrane-impermeable drugs.

ECT represents a promising minimally invasive treatment option for liver metastases. Its unique mechanism of action, favorable safety profile, and encouraging efficacy results make it an attractive alternative or complement to existing therapies.

Decision to treat liver metastases with ECT depends on various factors, including tumor characteristics, patient condition, device availability, and operator expertise, and should be considered in the context of a multidisciplinary tumor board discussion to tailor the most appropriate treatment for the patient’s needs, alone or in combination with other treatment modalities.

l  Electrochemotherapy has proven to be safe and effective in the treatment of colorectal liver metastases, with a durable response. It provides local tumor control that enables patients with unresectable metastases to receive further treatments.

l  Electrochemotherapy is a novel ablation technique combining chemotherapeutic agents with reversible cell membrane electroporation. Previous experiences have shown its efficacy for cutaneous tumors. Its application for deep-seated malignancies is under investigation. Electrochemotherapy is a feasible and safe adjunct to open surgery for treatment of unresectable colorectal liver metastases

*Electrochemotherapy provides non-thermal ablation of cutaneous as well as deep seated tumors. Based on positive results of the treatment of colorectal liver metastases. Electrochemotherapy is predominantly applicable in patients with impaired liver function due to liver cirrhosis and/or with lesions where a high-risk operation is needed to achieve curative intent, given the intra/perioperative risk for high morbidity and mortality.

*Electrochemotherapy is a local ablative therapy that increases the cytotoxicity of either bleomycin or cisplatin by applying electric pulses (electroporation) to tumors. It has already been widely used throughout Europe for the treatment of various types of human and veterinary cutaneous tumors, with an objective response rate ranging from 70%-90%, depending on the tumor histotype. Recently, electrochemotherapy was introduced for the treatment of primary liver tumors, such as hepatocellular carcinoma (HCC). The complete response rate was 85% per treated lesion, with a durable response. Therefore, electrochemotherapy could become a treatment of choice for HCC, especially after achieving a transition from an open surgery approach to a percutaneous approach that uses dedicated electrodes. Electrochemotherapy elicits a local immune response and can be considered an in situ vaccination. HCC, among others, is a potentially immunogenic tumor; thus, electrochemotherapy could boost adjuvant immunotherapy to achieve a better and longer-lasting antitumor response. Therefore, therapeutic strategies that combine electrochemotherapy with immune checkpoint inhibitors or adjuvant treatment with cytokines are indicated for HCC. Immunogene therapy using electroporation as a delivery system for plasmid DNA coding for interleukin-12 is a highly promising approach. This electroporation approach has shown efficacy in preclinical settings and veterinary oncology and is awaiting translation for the treatment of liver tumors, i.e., HCC.

*Above mentioned should be referred to.

Author Response

Comment:

It is an interesting manuscript about “Electrochemotherapy for Colorectal Liver Metastasis: what interventional radiologists need to know”.

*My concern is determined in the following points.

A potential alternative nonsurgical intervention such as electrochemotherapy (ECT) in treatment of CRC oligometastatic liver disease. ECT has been largely used for the treatment of cutaneous and subcutaneous lesions, while its visceral use is currently a novel approach. ECT consists in the administration of intravenous anticancer drugs, followed by the application of intralesional electrode-needles which release localized electrical pulses to induce electroporation, a process that transiently increases cell membrane permeability, thereby facilitating the intracellular delivery of otherwise membrane-impermeable drugs.

ECT represents a promising minimally invasive treatment option for liver metastases. Its unique mechanism of action, favorable safety profile, and encouraging efficacy results make it an attractive alternative or complement to existing therapies.

Decision to treat liver metastases with ECT depends on various factors, including tumor characteristics, patient condition, device availability, and operator expertise, and should be considered in the context of a multidisciplinary tumor board discussion to tailor the most appropriate treatment for the patient’s needs, alone or in combination with other treatment modalities.

l  Electrochemotherapy has proven to be safe and effective in the treatment of colorectal liver metastases, with a durable response. It provides local tumor control that enables patients with unresectable metastases to receive further treatments.

l  Electrochemotherapy is a novel ablation technique combining chemotherapeutic agents with reversible cell membrane electroporation. Previous experiences have shown its efficacy for cutaneous tumors. Its application for deep-seated malignancies is under investigation. Electrochemotherapy is a feasible and safe adjunct to open surgery for treatment of unresectable colorectal liver metastases

*Electrochemotherapy provides non-thermal ablation of cutaneous as well as deep seated tumors. Based on positive results of the treatment of colorectal liver metastases. Electrochemotherapy is predominantly applicable in patients with impaired liver function due to liver cirrhosis and/or with lesions where a high-risk operation is needed to achieve curative intent, given the intra/perioperative risk for high morbidity and mortality.

*Electrochemotherapy is a local ablative therapy that increases the cytotoxicity of either bleomycin or cisplatin by applying electric pulses (electroporation) to tumors. It has already been widely used throughout Europe for the treatment of various types of human and veterinary cutaneous tumors, with an objective response rate ranging from 70%-90%, depending on the tumor histotype. Recently, electrochemotherapy was introduced for the treatment of primary liver tumors, such as hepatocellular carcinoma (HCC). The complete response rate was 85% per treated lesion, with a durable response. Therefore, electrochemotherapy could become a treatment of choice for HCC, especially after achieving a transition from an open surgery approach to a percutaneous approach that uses dedicated electrodes. Electrochemotherapy elicits a local immune response and can be considered an in situ vaccination. HCC, among others, is a potentially immunogenic tumor; thus, electrochemotherapy could boost adjuvant immunotherapy to achieve a better and longer-lasting antitumor response. Therefore, therapeutic strategies that combine electrochemotherapy with immune checkpoint inhibitors or adjuvant treatment with cytokines are indicated for HCC. Immunogene therapy using electroporation as a delivery system for plasmid DNA coding for interleukin-12 is a highly promising approach. This electroporation approach has shown efficacy in preclinical settings and veterinary oncology and is awaiting translation for the treatment of liver tumors, i.e., HCC.

Reply:

Dear revisor, thank you for your considerations. We agree with them.

Reviewer 3 Report

Comments and Suggestions for Authors

The aim of this report is well presented. Using ECT is certainly a new approach fighting against colorectal liver metastasis, a clinical problem yet not fully resolved.

The different other approaches are well presented, references are complete and allow for additional information.

Perhaps the authors can include a para on prevention of colorectal carcinoma and resulting metastatic dissemination.

First, individuals should reduce potential risk factors, and second endoscopic approaches to search for adenomas and remove the polyps are strongly recommended.

Author Response

Comment:

The aim of this report is well presented. Using ECT is certainly a new approach fighting against colorectal liver metastasis, a clinical problem yet not fully resolved.

The different other approaches are well presented, references are complete and allow for additional information.

Perhaps the authors can include a para on prevention of colorectal carcinoma and resulting metastatic dissemination.

First, individuals should reduce potential risk factors, and second endoscopic approaches to search for adenomas and remove the polyps are strongly recommended.

Reply:

Dear reviewer, thanks for your comment and appreciation of the paper. We believe that the inclusion of a paragraph regarding the prevention of colorectal carcinoma would not be focused on the topic, considering that the purpose of the work is to describe an innovative therapeutic method that is still rarely used, and the need to raise awareness of it among physicians.

Reviewer 4 Report

Comments and Suggestions for Authors

In livers-3381809, Posa et al discuss electrochemotherapy for the management of colorectal lver metastasis. The topic of this review manuscript is interesting and fits well the scope of Livers. The reviewer feel it need extensive amendments  before it can be proceed. 

(1) As a review, the presentation is really so so. Why there is only text but no table and figure in this review? The presentation needs to be polished. Color figures / diagrams are encouraged. 

(2) Any existing meta-analysis?

(3) Any cost-effectiveness data?

(4) Can such approach be extended to other type of cancer?

Author Response

Comment:

In livers-3381809, Posa et al discuss electrochemotherapy for the management of colorectal liver metastasis. The topic of this review manuscript is interesting and fits well the scope of Livers. The reviewer feel it need extensive amendments  before it can be proceed.

(1) As a review, the presentation is really so so. Why there is only text but no table and figure in this review? The presentation needs to be polished. Color figures / diagrams are encouraged.

(2) Any existing meta-analysis?

(3) Any cost-effectiveness data?

(4) Can such approach be extended to other type of cancer?

Reply:

Dear author, thank you for your precious comments.

1- Draft has been polished, figures were provided in the final draft as requested.

2- A meta-analysis is not available in literature due to the scarcity and heterogeneity of data; there is only a systematic review by Barbieri et al. cited in the article lacking a meta-analysis for this reason.  

3-A cost-effectiveness analysis is not available in literature and it might be difficult to achieve considering the heterogeneity of the techniques and materials employed and the different courts of patients to be investigated.

4-As mentioned in the draft, ECT can be applied in various solid tumors (see lines 71, 72); however, the narrative focuses on mCRC liver metastases.

Round 2

Reviewer 4 Report

Comments and Suggestions for Authors

The manuscript has been improved and the reviewer has no objection to pass it.